# Impact of Developing Dialysis-Requiring Acute Kidney Injury on Long-Term Mortality in Cancer Patients with Septic Shock

**DOI:** 10.3390/cancers15143619

**Published:** 2023-07-14

**Authors:** June-Sung Kim, Ye-Jee Kim, Youn-Jung Kim, Won Young Kim

**Affiliations:** 1Department of Emergency Medicine, University of Ulsan College of Medicine, Asan Medical Center, Seoul 05505, Republic of Korea; jsmeet09@gmail.com; 2Department of Clinical Epidemiology and Biostatistics, University of Ulsan College of Medicine, Asan Medical Center, Seoul 05505, Republic of Korea; kimyejee@amc.seoul.kr

**Keywords:** cancer, sepsis, septic shock, mortality, acute kidney injury

## Abstract

**Simple Summary:**

Although studies have evaluated the association between sepsis-related acute kidney injury and mortality, less is known about the epidemiology and course in cancer patients. This population-based cohort study using data from the Nation Health Insurance Service of Korea aimed to determine whether dialysis-requiring acute kidney injury could be related to an increase in short-term and long-term negative outcomes. All cancer patients admitted to a hospital via the emergency department with septic shock from 2009 to 2017 were included. Dialysis-requiring septic AKI occurred in 13% of adult cancer patients with septic shock and was associated with male sex, hematologic cancers, and comorbidities. Moreover, dialysis-requiring septic acute kidney injury was significantly associated with increased long-term mortality in the cancer patients, suggesting that the prevention of acute kidney injury, particularly in male hematologic cancer patients, should be emphasized.

**Abstract:**

(1) Background: Considering recent advances in both cancer and sepsis management, we chose to evaluate the associated factors for occurrence of septic acute kidney injury in cancer patients using a nationwide population-based cohort data. (2) Methods: Using data from the National Health Insurance Service of Korea, adult cancer patients who presented to emergency departments with septic shock from 2009 to 2017 were analyzed. A Cox-proportional hazard model was conducted to evaluate the clinical effect of sepsis-related acute kidney injury requiring dialysis. (3) Results: Among 42,477 adult cancer patients with septic shock, dialysis-requiring acute kidney injury occurred in 5449 (12.8%). Recovery from dialysis within 30 days was 77.9% and, overall, 30-day and 2-year mortality rates were 52.1% and 85.1%, respectively. Oncologic patients with dialysis-requiring acute kidney injury frequently occurred in males and patients with hematologic cancer. A multivariate Cox-proportional hazard model showed that dialysis-requiring acute kidney injury had the highest adjusted hazard ratio of 1.353 (95% confidence interval 1.313–1.395) for 2-year mortality. (4) Conclusions: Dialysis-requiring septic acute kidney injury did not occur commonly. However, it had a significant association with increased long-term mortality, which suggests emphasis should be placed on the prevention of acute kidney injury, particularly in male hematologic cancer patients.

## 1. Introduction

Although advances in the diagnosis and treatment of cancer have improved survival outcomes, the numbers of cancer patients with critical illnesses have increased in association with the longer duration of cancers [1,2]. Septic shock is one of the most common life-threatening complications in cancer patients, leading to multi-organ failure with a high mortality rate [3,4,5]. Sepsis, the most frequent cause of acute kidney injury (AKI), is associated not only with higher in-hospital mortality rate but also with the later development of chronic kidney disease and end-stage renal disease, and with long-term increases in medical costs and risk of death [6]. These adverse impacts are even more evident in patients with severe AKI requiring dialysis, with the risk of AKI and the requirement for dialysis found to be higher in critically ill patients with cancer than without [7,8]. This higher incidence of AKI in cancer patients is likely due to their exposure to nephrotoxic drugs and radiation [9]. In addition, increased age and higher incidences of comorbidities, including cardiovascular disease and chronic kidney disease, may contribute to the susceptibility of cancer patients to kidney damage [10,11].

Although studies have evaluated the association between sepsis-related AKI and mortality, less is known about the epidemiology and course of AKI in cancer patients, as most evaluations have included small sample sizes and individual types of cancers [12]. Moreover, the association between sepsis-related AKI and long-term mortality is particularly unclear, and may have been altered by recent improvements in the care of cancer patients and protocol-based resuscitation bundle therapy for septic shock. We postulated that the occurrence of dialysis-requiring septic AKI among cancer patients could impact negative outcomes. Using nationwide population-based cohort data with withdrawal of life-sustaining therapy excluded, the present study evaluated factors associated with the development of septic AKI in cancer patients with septic shock and assessed the effects of dialysis-requiring AKI on long-term mortality.

## 2. Materials and Methods

### 2.1. Study Design

This population-based cohort study used data from the Korean National Health Information Database (NHID) that were collected between 2009 and 2017 and released in 2019. The Korean National Health Insurance Service (NHIS) has been the single insurer covering all South Korean citizens since the enactment of the Medical Insurance Act in 1963 [13]. The Korean NHIS database includes all claims data, such as patient demographics, drug prescriptions, ICD-10 diagnostic codes, insurance coverage and payments, patient deductibles, and treatment details [13]. However, laboratory and radiologic data were not available from the NHID. This study was approved by the Institutional Review Board of the Asan Medical Center (study number: 2019-0743), which waived the requirement for informed consent due to de-identification of claims data. All methods were performed in accordance with the relevant guidelines and regulations.

### 2.2. Patients and Data Collection

All cancer patients admitted to a hospital via the emergency department (ED) and who fulfilled the clinical surveillance definition of septic shock were included. Patients were selected if they had an ICD-10 diagnosis code for cancer and a registration code for a rare incurable disease (V193, V027) within the preceding 90 days of hospitalization for septic shock. In South Korea, the NHIS cancer registration program for financial support and scrutiny assigned cancer diagnosis codes to Korean cancer patients as either a principal or secondary diagnosis [14]. The accuracy of screening cancer patients through ICD-10 codes (C00–C97) and cancer registration codes (V193, V027) has been estimated to be 98.2% [15]. Cancer patients with septic shock were identified using a clinical surveillance definition of septic shock, based on concurrent vasopressors, antibiotics, and blood cultures [16,17]. The Third International Consensus Definition for Sepsis and Septic Shock (Sepsis-3) has defined septic shock as a “life-threatening organ dysfunction caused by a dysregulated host response to infection, requiring vasopressor therapy, and know elevated lactate level.” [18]. Thus, of patients with a blood culture order and concomitant administration of intravenous antibiotics for suspected infection, those who received any type of vasopressor, including dopamine, norepinephrine, epinephrine, vasopressin, or phenylephrine, were defined as having septic shock. CCI for measuring the burden of underlying illnesses was automatically calculated in NHID data and extracted. 

Because laboratory data, including creatinine level and glomerular filtration rate, were not available from the NHID, dialysis-requiring AKI in oncologic patients with septic shock was defined as a requirement for continuous RRT or hemodialysis, as determined by diagnostic codes, at the time of admission, whereas patients who had been previously diagnosed with end-stage renal disease and received RRT within one year were excluded [19]. Recovery from dialysis-dependent AKI within one month was determined based on diagnostic codes for RRT or hemo- or peritoneal dialysis. If patients visited more than once for septic shock, data collected at the first admission were included.

### 2.3. Statistical Analysis

Descriptive analyses were performed to compare baseline characteristics in patients who did and did not require dialysis for septic AKI. Categorical variables are presented as numbers and percentages and compared using Chi-square tests, whereas non-normally distributed continuous variables are presented as medians and interquartile ranges (IQR) and compared using Mann–Whitney U tests. Independent risk factors for developing septic AKI were determined by backward stepwise multivariate logistic regression analyses with statistically different parameters (*p* values < 0.1) in univariate analysis. A Cox-proportional hazards model with multivariable adjustment was utilized to compare survivors and non-survivors within two years. All statistical analyses were performed using Enterprise Guide version 7.1 (SAS Institute Inc., Cary, NC, USA), with *p* values < 0.05 considered statistically significant.

## 3. Results

### 3.1. Study Design and Population

Between 2009 and 2017, 322,142 adult patients with septic shock were admitted to hospitals in South Korea through the ED (Figure 1). Of these, 43,466 patients (13.5%) had International Classification of Disease 10th edition (ICD-10) codes for active cancer. After excluding the 989 patients who had undergone hemo- or peritoneal dialysis within one year prior to their ED visits, 42,477 cancer patients with septic shock were included. Of these, 5449 patients (12.8%) had dialysis-requiring AKI during hospitalization, with 4244 (77.9%) of the latter recovering from dialysis within 30 days.

### 3.2. Baseline Characteristics of the Study Population

Table 1 shows the baseline demographic and clinical characteristics of the overall study population and of patients who did and did not develop dialysis-requiring septic AKI. The overall patient population consisted of 27,419 (64.5%) men and 15,058 (35.5%) women of median age 69.0 years. Patients with dialysis-requiring septic AKI were younger (68.0 vs. 69.0 years) and had lower Charlson Comorbidity Index (CCI) scores (5.0 vs. 6.0) than patients who did not require dialysis. Hypertension, diabetes mellitus, and liver cirrhosis were more frequent in patients who did require dialysis than those who did not. Moreover, hematologic cancers were more frequent in patients who were dialysis-dependent than those who were not (14.1% vs. 5.1%).

### 3.3. Factors Predicting the Occurrence of Dialysis-Requiring Septic AKI

Table 2 shows the results of univariate and multivariate logistic regression analyses of factors predicting the occurrence of dialysis-requiring septic AKI. Female sex (adjusted odds ratio [OR]: 0.848, 95% confidence interval [CI]: 0.797–0.903) and older age (adjusted OR: 0.989, 95% CI: 0.987–0.991) were factors associated with a reduced likelihood of developing septic AKI. All comorbidities, except for chronic obstructive pulmonary disease (adjusted OR: 0.840, 95% CI: 0.759–0.928), were independent risk factors for developing dialysis-requiring septic AKI, as was hematologic compared with solid cancers (adjusted OR: 2.652, 95% OR: 2.414–2.911).

### 3.4. Characteristics of the Two-Year Non-Survivors and Survivors

The one-month all-cause mortality rate in cancer patients with septic shock was 51.9% (*n* = 22,065) and the two-year mortality rate was 85.1% (*n* = 36,161). Table 3 shows the characteristics of the two-year survivors and non-survivors. Dialysis requirement during hospitalization was more frequent in non-survivors than in survivors. Two-year survivors were younger (64.0 vs. 69.0 years) and had a lower CCI (4.0 vs. 6.0) than non-survivors, and the percentage of women (39.7% vs. 34.7%) was higher in two-year survivors. All comorbidities were less common in survivors, but the percentages of patients with solid and hematologic cancers were similar in the two groups. Similar results were observed in comparisons of one-month survivors and non-survivors (see Appendix A).

### 3.5. Cox-Proportional Hazard Analyses of Factors Associated with Two-Year Mortality

Table 4 shows the results of univariate and multivariate Cox-proportional hazard analyses of factors associated with two-year mortality. Older age, higher CCI, and hematologic cancer were associated with higher two-year mortality rate. Notably, the occurrence of the dialysis-requiring AKI was an independent risk factor for two-year mortality (hazard ratio [HR]: 1.353, 95% CI, 1.313–1.395, *p* < 0.001). Similar results were observed on univariate and multivariate Cox-proportional hazard analyses of factors associated with one-month mortality (see Appendix A). Kaplan–Meier survivor analyses also showed that two-year (Figure 2) and one-month (see Appendix A) survival rates were lower in cancer patients with dialysis-requiring septic AKI than without. 

## 4. Discussion

This nationwide study found that, of 42,477 adult cancer patients admitted to hospital with septic shock, about 13% experienced dialysis-requiring AKI during hospitalization. Factors associated with dialysis-requiring septic AKI included comorbidities, such as hypertension, diabetes, congestive heart failure, and liver cirrhosis, as well as hematologic cancer. Multivariate Cox-proportional hazard analysis showed that dialysis-requiring AKI had the highest adjusted HR (1.35; 95% CI: 1.31–1.40) for two-year mortality among cancer patients with septic shock.

The incidence of AKI among cancer patients appears to be increasing due to aggressive treatments and effective critical care. Nationwide studies have reported that the rate of any AKI among oncology patients requiring renal replacement therapy (RRT) ranged from 10% to 50% [20,21]. These differences may be due to differences in the numbers and ethnicities of patients included in these studies, the definition of AKI, the severity of disease, and the stage of cancer. The incidence of AKI was found to be higher in cancer patients with critical illnesses than in non-cancer patients with similar severity of diseases [22,23]. Although relatively little is known about the epidemiology of septic AKI in cancer, the present study showed that about 13% of cancer patients with septic shock had dialysis-requiring AKI during hospitalization. The lower incidence in this study than in previous studies may have been due to our exclusion of patients who had previously received RRT before ED arrival, as prior RRT may be a risk factor for RRT during hospitalization [24]. This study, however, was the largest population-based study of septic AKI among cancer patients in South Korea, and may provide evidence on the epidemiology of critically ill cancer patients with a dialysis dependency.

This study also found that male sex, comorbidities, and hematologic cancers were factors associated with more frequent RRT. In contrast to results showing that older age was a risk factor for the development of AKI requiring RRT [25], the present study found that patients with dialysis-requiring AKI were of lower median age than patients not requiring dialysis. One possibility may have been that older patients declined invasive procedures, including RRT. Although we did not determine the numbers of patients with do-not-resuscitate orders, older patients were more likely to decline aggressive intubation, ICU admission, and RRT.

Furthermore, the present study found that the relative incidence of AKI was higher in patients with hematologic than solid cancers, confirming results showing that non-septic patients with hematologic malignancies, such as leukemia, lymphoma, and multiple myeloma, were at highest risk for AKI development [26]. Compared with patients with solid cancers, patients with hematologic malignances were more susceptible to infection, had more severe disease, were more frequently treated with nephrotoxins, and more frequently had metabolic disturbances [27].

Our data showed that the overall two-year mortality rate was 85.1% in cancer patients with dialysis-requiring septic AKI. Although several retrospective studies evaluated the association between septic AKI requiring RRT and short-term outcomes, it may be difficult to determine the contribution of RRT-dependent septic AKI to long-term mortality, as cancer itself is a highly burdensome underlying disease. Using nationwide population data with withdrawal of life-sustaining therapy being excluded, the present study found that dialysis-dependent septic AKI was an independent risk factor for long-term mortality after adjustment for known confounders. Mortality rates one year after AKI were found to be higher in cancer patients than in patients with other chronic illnesses [28]. Moreover, a trial that included only critically ill patients with dialysis-requiring AKI showed that infection-related factors, such as sepsis, were the most frequent causes of death [29]. The negative impact of septic AKI on clinical outcomes in cancer patients was multifactorial. For example, AKI can negatively influence the effectiveness of current or future chemotherapeutic regimens, as well as increasing their side effects or altering their pharmacodynamics and pharmacokinetics. Moreover, AKI would preclude patients from participating in potentially beneficial clinical trials.

This study had both strengths and limitations. One important strength was that we analyzed a recent national database with high coverage of the population (97%), and this was the largest study regarding AKI among cancer patients on long-term mortality. This may provide a more comprehensive understanding of cancer patients with septic shock. One limitation, however, was that the NHIS database did not include detailed clinical and laboratory data, which may have had confounding effects on patient outcomes. Parameters related with organ dysfunctions, such as serum platelet, creatinine, and bilirubin, might reflect severity of acute insult for septic patients. Furthermore, the NHIS database did not include specific cancer stage, performance status, or treatment setting that could influence long-term mortality. Finally, this study lacked detailed information on septic shock treatment, such as the timing of fluids, antibiotics, and vasopressor administration, all of which could have affected clinical outcomes. Well-designed multicenter prospective studies with large sample sizes will be needed to identify detailed risk factors and prognoses of septic AKI among patients with active cancers.

## 5. Conclusions

In conclusion, dialysis-requiring septic AKI occurred in 13% of adult cancer patients with septic shock and was associated with male sex, hematologic cancers, and comorbidities. Moreover, dialysis-requiring septic AKI was significantly associated with increased long-term mortality in the cancer patients. Prospective studies are needed to investigate methods to prevent and properly manage dialysis-requiring septic AKI in adult cancer patients with septic shock.

## Figures and Tables

**Figure 1 cancers-15-03619-f001:**
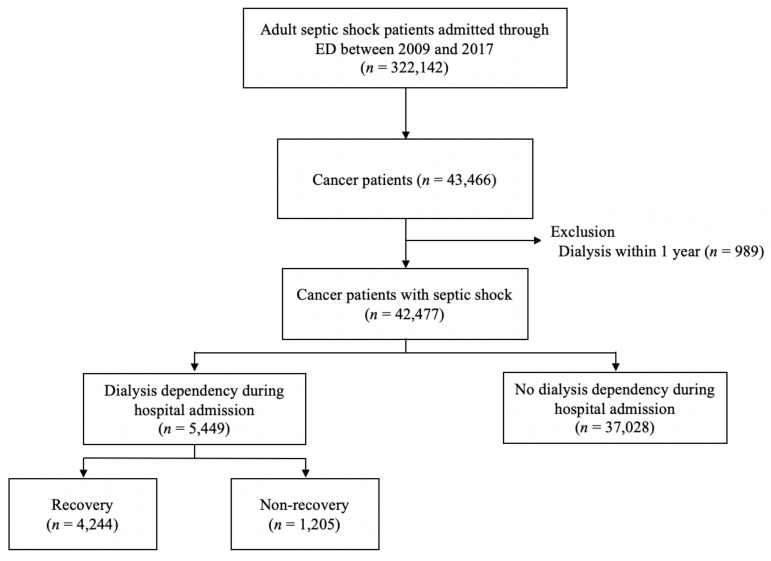
Patient flow chart.

**Figure 2 cancers-15-03619-f002:**
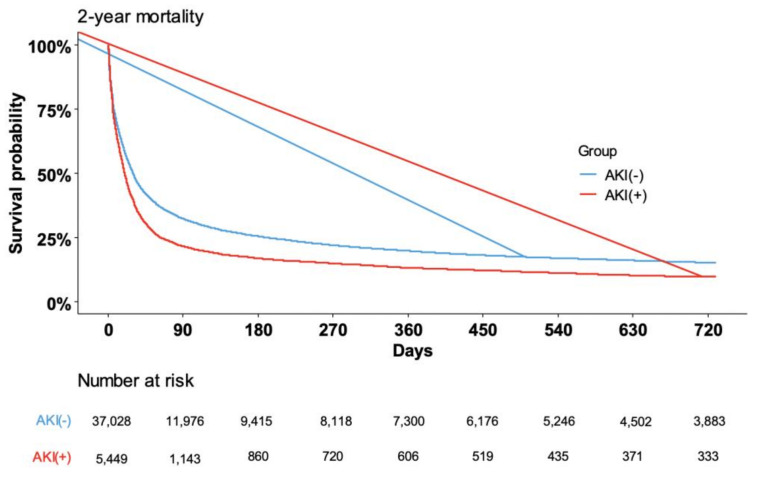
Kaplan-Myer analysis of two-year mortality in cancer patients with septic shock who were and were not dependent on dialysis during hospitalization.

**Table 1 cancers-15-03619-t001:** Demographic and clinical characteristics of cancer patients with and without dialysis-dependent AKI.

Characteristics	Total (*n* = 42,477)	Dialysis-Independent (*n* = 37,028)	Dialysis-Dependent (*n* = 5449)	*p*-Value
Female	15,058 (35.5)	13,282 (35.9)	1776 (32.6)	<0.01
Age (year) ^1^	69.0 (59.0–76.0)	69.0 (59.0–77.0)	68.0 (58.0–75.0)	<0.01
CCI ^1^	6.0 (3.0–9.0)	6.0 (3.0–9.0)	5.0 (3.0–8.0)	<0.01
Underlying disease				
Hypertension	22,338 (52.6)	19,191 (51.8)	3147 (57.8)	<0.01
Diabetes mellitus	16,321 (38.4)	13,993 (37.8)	2328 (42.7)	<0.01
Congestive heart failure	5475 (12.9)	4663 (15.6)	812 (14.9)	<0.01
COPD	4903 (11.5)	4397 (11.9)	506 (9.3)	<0.01
Liver cirrhosis	4823 (11.4)	4030 (10.9)	793 (14.6)	<0.01
Cancer type				
Solid ^2^	39,811 (93.7)	35,129 (94.9)	4682 (85.9)	<0.01
Hematologic ^3^	2666 (6.3)	1899 (5.13)	767 (14.1)	<0.01

Unless otherwise indicated, data are presented as numbers (%). ^1^ Data presented as the median (interquartile range). ^2^ Including cancers of the brain, lung, liver, colon, stomach, gall bladder, and pancreas and lymphoma. ^3^ Including multiple myeloma and leukemia. Abbreviations: AKI, acute kidney injury; CCI, Charlson Comorbidity Index; COPD, chronic obstructive pulmonary disease.

**Table 2 cancers-15-03619-t002:** Univariate and multivariate logistic regression analyses of factors predicting dialysis-dependent septic AKI.

Variables	Univariable			Multivariable		
	OR	95% CI	*p*	Adjusted OR	95% CI	*p*
Female	0.86	0.81–0.92	<0.01	0.85	0.80–0.90	<0.01
Age	0.99	0.99–0.99	<0.01	0.99	0.99–0.99	<0.01
Hypertension	1.27	1.20–1.35	<0.01	1.39	1.30–1.48	<0.01
Diabetes	1.23	1.20–1.61	<0.01	1.36	1.27–1.45	<0.01
CHF	1.22	1.121–1.32	<0.01	1.23	1.13–1.34	<0.01
Chronic pulmonary diseases	0.76	0.69–0.84	<0.01	0.84	0.76–0.93	<0.01
LC	1.40	1.28–1.51	<0.01	1.51	1.38–1.64	<0.01
CCI	0.95	0.94–0.96	<0.01	0.94	0.93–0.94	<0.01
Solid cancer ^1^	Reference			Reference		
Hematologic cancer ^2^	3.03	2.77–3.31	<0.01	2.65	2.41–2.91	<0.01

^1^ Including cancers of the brain, lung, liver, colon, stomach, gall bladder, and pancreas and lymphoma. ^2^ Including multiple myeloma and leukemia. Abbreviations: OR, odds ratio; CI, confidence interval; CHF, congestive heart failure; LC, liver cirrhosis; CCI, Charlson Comorbidity Index.

**Table 3 cancers-15-03619-t003:** Demographic and clinical characteristics of two-year survivors and non-survivors.

Characteristics	Total (*n* = 42,477)	2-Year Survivors (*n* = 6316)	2-Year Non-Survivors (*n* = 36,161)	*p* Value
Dialysis	5449 (12.8)	504 (8.0)	4945 (13.7)	<0.01
Female	15,058 (35.5)	2505 (39.7)	12,553 (34.7)	<0.01
Age (year) ^1^	69.0 (59.0–76.0)	64.0 (55.0–73.0)	69.0 (60.0–77.0)	<0.01
CCI (continuous) ^1^	6.0 (3.0–9.0)	4.0 (2.0–7.0)	6.0 (3.0–10.0)	<0.01
Underlying disease				
Hypertension	22,338 (52.6)	2989 (47.3)	19,349 (53.5)	<0.01
Diabetes mellitus	16,321 (38.4)	2041 (32.3)	14,280 (39.5)	<0.01
Congestive heart failure	5475 (12.9)	645 (10.2)	4830 (13.4)	<0.01
COPD	4903 (11.5)	458 (7.3)	4445 (12.3)	<0.01
Chronic kidney disease	2083 (4.9)	278 (4.4)	1805 (5.0)	<0.01
Liver cirrhosis	4823 (11.4)	598 (9.5)	4225 (11.7)	<0.01
Cancer type				
Solid ^2^	39,811 (93.7)	5912 (93.6)	33,899 (93.7)	0.67
Hematologic ^3^	2666 (6.3)	404 (6.4)	2262 (6.3)	0.67

Unless otherwise indicated, data are presented as numbers (%). ^1^ Data presented as median (interquartile range). ^2^ Including cancers of the brain, lung, liver, colon, stomach, gall bladder, and pancreas and lymphoma. ^3^ Hematologic cancer included multiple myeloma and leukemia. Abbreviations: CCI, Charlson Comorbidity Index; COPD, chronic obstructive pulmonary disease.

**Table 4 cancers-15-03619-t004:** Univariate and multivariate Cox-proportional hazard analyses of factors predicting two-year mortality.

Variables	Univariable			Multivariable		
	HR	95% CI	*p* Value	Adjusted HR	95% CI	*p* Value
Dialysis	1.31	1.27–1.35	<0.01	1.35	1.31–1.40	<0.01
Female	0.91	0.89–0.93	<0.01	0.94	0.92–0.96	<0.01
Age	1.01	1.01–1.01	<0.01	1.01	1.01–1.01	<0.01
Hypertension	1.09	1.07–1.12	<0.01	0.94	0.92–0.97	<0.01
Diabetes	1.11	1.09–1.4	<0.01	0.90	0.88–0.93	<0.01
CHF	1.13	1.10–1.14	<0.01	0.98	0.95–1.01	0.24
COPD	1.25	1.21–1.29	<0.01	1.09	1.06–1.13	<0.01
LC	1.23	1.19–1.27	<0.01	1.18	1.14–1.22	<0.01
CCI	1.06	1.06–1.06	<0.01	1.07	1.06–1.07	<0.01
Solid cancer ^1^	Reference			Reference		
Hematologic cancer ^2^	1.01	0.97–1.05	0.70	1.19	1.14–1.25	<0.01

^1^ Including cancers of the brain, lung, liver, colon, stomach, gall bladder, and pancreas and lymphoma. ^2^ Including multiple myeloma and leukemia. Abbreviations: HR, hazard ratio; CI, confidence interval; CHF, congestive heart failure; COPD, chronic obstructive pulmonary disease; LC, liver cirrhosis; CCI, Charlson Comorbidity Index.

## Data Availability

No new data were created or analyzed in this study. Data sharing is not applicable to this article.

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
