# Peer review of "Impact of Developing Dialysis-Requiring Acute Kidney Injury on Long-Term Mortality in Cancer Patients with Septic Shock"

_cancers, 2023, doi:10.3390/cancers15143619_

Round 1
Reviewer 1 Report
The authors present a review paper on acute kidney injury in cancer patients who have suffered sepsis.
It is noteworthy that the new treatments for cancer and the survival that is achieved may pose problems that a few years ago were not assessed in these patients.
For this evaluation, the authors have carried out a retrospective study based on National Health of Korea, during the period from 2009 to 2017. This led to an analysis of more than 40,000 patients, a fact that must be highlighted.
The clear conclusions reveal that septic acute kidney injury requiring dialysis was significantly associated with increased long-term mortality that emphasizes the prevention of acute kidney injury particularly in male patients with haematological cancer.
I think it is an original study, well done and that presents clear clinical conclusions
Author Response
The authors present a review paper on acute kidney injury in cancer patients who have suffered sepsis.
It is noteworthy that the new treatments for cancer and the survival that is achieved may pose problems that a few years ago were not assessed in these patients.
For this evaluation, the authors have carried out a retrospective study based on National Health of Korea, during the period from 2009 to 2017. This led to an analysis of more than 40,000 patients, a fact that must be highlighted.
The clear conclusions reveal that septic acute kidney injury requiring dialysis was significantly associated with increased long-term mortality that emphasizes the prevention of acute kidney injury particularly in male patients with haematological cancer.
I think it is an original study, well done and that presents clear clinical conclusions
Response> Thanks for the generous comments.
Reviewer 2 Report
-
Kim et al, have penned a comprehensive nationwide study with a large cohort of cancer patients suffering from AKI-induced septic shock, emphasizing the epidemiology of the disease and the risk factors involved in the progression of the disease that compromise mortality in the cancer patients. The results are intriguing, but as an active cancer biologist, I have the following concerns or questions that, if answered, would strengthen the study in the long run:
1. The hereditary risk factors involved in the progression of cancer in AKI patients need to be investigated, as some types of hereditary kidney cancer, such as VHL disease, can be more or less aggressive than sporadic kidney cancer.
2. The study requires significant clinical or laboratory data to corroborate its valuable findings.
3. The study can be further expanded with a much larger sample size by incorporating data from worldwide databases such as NIH/NCI and may be compared with Korean NHIS to seek similarities or differences amongst them.
4. The treatment for hereditary kidney cancer may differ from treatments for sporadic kidney cancer, which should be one of the imperative subjects to be considered to enhance the physiological significance of the study.
Author Response
Kim et al, have penned a comprehensive nationwide study with a large cohort of cancer patients suffering from AKI-induced septic shock, emphasizing the epidemiology of the disease and the risk factors involved in the progression of the disease that compromise mortality in the cancer patients. The results are intriguing, but as an active cancer biologist, I have the following concerns or questions that, if answered, would strengthen the study in the long run:
- The hereditary risk factors involved in the progression of cancer in AKI patients need to be investigated, as some types of hereditary kidney cancer, such as VHL disease, can be more or less aggressive than sporadic kidney cancer.
Response> Thanks for the valuable comments. We totally agreed with the reviewer’s opinion that hereditary risk factors including VHL disease could impact the development of dialysis-required septic AKI. However, because of the retrospective design and the characteristics of the nationwide population data, we could not collect such data. Further studies will be needed to clarify this suggestion.
- The study requires significant clinical or laboratory data to corroborate its valuable findings.
Response> We agreed with the reviewer’s concern that the clinical and laboratory data, including serum lactate, creatinine, and bilirubin, for assessing the severity of sepsis could reveal more information about the prognosis of septic AKI. Regretfully, the Korean National Health Information Database did not include such data and we already mentioned these limitations in the manuscript. The well-designed, multicenter prospective study can make up for this limitation in the future.
- The study can be further expanded with a much larger sample size by incorporating data from worldwide databases such as NIH/NCI and may be compared with Korean NHIS to seek similarities or differences among them.
Response> Thanks for the great suggestions. We will try to analyze with a much larger sample and to reveal more in-depth about cancer patients.
- The treatment for hereditary kidney cancer may differ from treatments for sporadic kidney cancer, which should be one of the imperative subjects to be considered to enhance the physiological significance of the study.
Response> Because we did not have data about patients with hereditary kidney cancer, we could not evaluate the impact of the difference between diseases.